# Mesoporous CuO Prepared in a Natural Deep Eutectic Solvent Medium for Effective Photodegradation of Rhodamine B

**DOI:** 10.3390/molecules28145554

**Published:** 2023-07-20

**Authors:** Sheli Zhang, Yuerong Yan, Wei Hu, Yunchang Fan

**Affiliations:** 1College of Science and Technology, Jiaozuo Teachers College, Jiaozuo 454000, China; 1295006035@jzsz.edu.cn (S.Z.); 1295002321@jzsz.edu.cn (Y.Y.); 2College of Chemistry and Chemical Engineering, Henan Polytechnic University, Jiaozuo 454003, China; huwei@home.hpu.edu.cn

**Keywords:** natural deep eutectic solvents, photocatalytic degradation, copper oxide, rhodamine B, photoexcited holes

## Abstract

Metal oxide nanoparticles (NPs) have been widely used as catalysts in the chemical industry, but their preparation is usually limited by strict conditions such as high temperature, elevated pressure, and the use of volatile and highly toxic organic solvents. To solve this problem, this work developed an environmentally benign method using green solvents, i.e., natural deep eutectic solvents (NADESs), as a reaction medium to prepare copper oxide (CuO) particles. The experimental results suggested that the synthesized CuO particles were sheet-like mesoporous NPs, and they exhibited excellent catalytic performance towards the photodegradation of rhodamine B (RhB) in the presence of potassium monopersulfate (PMS). The catalytic activity of the synthesized CuO NPs was better than that of the reported metal oxide-based catalysts. Reactive species such as photoexcited holes, superoxide radicals, and singlet oxygen were involved in the RhB degradation. These results indicated that NADESs are good media for the preparation of CuO NPs, and exhibit the potential for application to the preparation of other metal oxides.

## 1. Introduction

Cupric oxide (CuO), a *p*-type semiconductor, with a narrow band gap of about 1.7 eV has been widely used as a photocatalyst in the degradation of organic pollutants, because of its low cost and high efficiency in absorbing sunlight [1,2]. Generally, CuO particles with various structures are synthesized via solvothermal [1,3], ultrasound-assisted [4,5], and microwave assisted [6,7] methods. For example, Sun et al. prepared sheet-like CuO using a solvothermal method (reaction time, 12 h; reaction temperature, 80 °C) and the synthesized sheet-like CuO NPs could efficiently activate peroxydisulfate to degrade bisphenol A [1]. Rao et al. developed an ultrasound-assisted method to synthesize hierarchical CuO microstructures (ultrasonic power, 37 KHz; ultrasonic time, 3 h) and the morphology of the CuO could be controlled using different organic solvents as reaction media; the cotton-like CuO microstructures prepared in a water/methanol medium exhibited better photocatalytic performance in the degradation of Reactive Black 5 dye under visible light irradiation [4]. Bhattacharjee et al. synthesized 2D CuO nanoleaves using glutamic acid as a complexing and capping agent, CuSO_4_ as a precursor, and NaOH as a precipitation reagent under microwave irradiation (microwave irradiation time, thirty 10 s shots; frequency of microwave irradiation, 2450 MHz; power of microwave irradiation, 1350 W); the prepared 2D CuO nanoleaf proved to be efficient photocatalysts for the degradation of rose bengal and methyl violet 6B dyes [6].

Recently, CuO NPs were also applied as catalysts for the degradation of RhB [7,8]. Xiao et al. developed a Fenton-like catalyst by combining CuO with ascorbic acid (Vc) to activate H_2_O_2_ and degrade RhB. The removal efficiency of RhB reached 76% [7]. Liu and co-workers prepared oxygen vacancy-rich CuO NPs to construct a CuO/PMS system for the degradation of RhB. It was found that 98.0% RhB was degraded after 20 min in the presence of 0.12 g L^−1^ PMS and 0.3 g L^−1^ CuO under irradiation with a 300 W Xe lamp [8]. Although the CuO NPs exhibited excellent performance as photocatalysts in the degradation of RhB, the RhB degradation efficiency of the CuO/Vc/H_2_O_2_ system is only 76% [7]. Oxygen vacancy-rich CuO NPs exhibited a high RhB degradation efficiency, but an additional light source (Xe lamp) was required to irradiate the RhB degradation reaction [8]. These results suggested that the synthesis of CuO with enhanced catalytic activity is urgent. Additionally, the preparation of CuO is usually limited by strict experimental conditions (e.g., solvothermal and microwave-assisted techniques usually operate at high temperatures and elevated pressures [1,3,6,9]) and the use of toxic volatile organic solvents such as dimethylformamide, acetonitrile, and dimethyl sulfoxide [4]. Therefore, the development of a more straightforward, efficient, and sustainable technique of CuO synthesis using environmentally benign solvents under mild conditions is highly desirable. Recently, deep eutectic solvents (DESs), typically prepared by simply mixing a hydrogen bond acceptor (HBA) with a hydrogen bond donor (HBD), have received much attention for their excellent biocompatibility, chemical stability, and biodegradability. When DESs are derived from natural compounds, they are referred to as natural deep eutectic solvents (NADESs) [10,11]. As green alternatives to conventional solvents, DESs have been employed as reaction media for the synthesis of various functional materials, including noble metal nanomaterials, polymers, transition metal oxide nanomaterials, metal organic frameworks (MOFs) and covalent organic frameworks (COFs) [12,13,14,15]. In this context, the utilization of DES media for the preparation of CuO has also been explored; Shinde and co-workers used a DES composed of choline chloride and urea (CCU) as the reaction medium to prepare nanostructured CuO under stirring at 70 °C for 12 h, and the resulting CuO NPs exhibited noteworthy electrochemical performance [16]. Verma et al. also prepared chloride anion-doped CuO (Cl-CuO) using a CCU-based DES as a green solvent, and the prepared Cl-CuO NPs showed excellent photocatalytic performance for the degradation of 4-nitrophenol [17].

Although the utilization of CCU-based DESs to prepare CuO NPs has achieved exciting results, there remains one challenge. During the preparation process of CuO NPs, urea, one of the components of CCU-based DESs, will decompose into ammonia and isocyanic acid, which means that the DES structure will be destroyed, and the DES cannot be recycled [16]. The objectives of this work are therefore to investigate the feasibility of the use of a stable hydrophobic DES composed of *DL*-menthol (ML) and fenchyl alcohol (FA) (denoted ML-FA) as a reaction medium to prepare mesoporous CuO NPs and to systematically study their photocatalytic performance for the degradation of RhB.

## 2. Results and Discussion

### 2.1. The Reaction Mechanism of the Preparation of CuO NPs

In this work, a hydrophobic NADES, ML-FA (ML and FA act as HBA and HBD, respectively; its solubility in water is 48.1 mg (100 mL)^−1^ [18]), was used as a reaction medium to prepare CuO NPs (Figure 1). 1-(*n*-Butyl)imidazole (BIM) was added into Cu(NO_3_)_2_ aqueous solution to form the Cu(BIM)_4_(NO_3_)_2_ complex [19], followed by the addition of NADES to construct a two-phase system in which Cu(BIM)_4_(NO_3_)_2_ will transfer into the NADES phase from the water phase. After decanting the water phase, the NaOH aqueous solution was mixed with the NADES phase under stirring and heating, leading to the generation of CuO NPs at the water–NADES interface.

### 2.2. Characterization of the Synthesized CuO NPs

The structural characteristics of the CuO NPs synthesized under different conditions (Table 1) were investigated, and their X-ray diffraction (XRD) patterns illustrated in Figure 2 indicate that the diffraction peaks of the synthesized CuO NPs are in good accordance with those of monoclinic symmetry CuO (JCPDS No. 89-2529), thus confirming the purity of the CuONPs. The lattice parameters of the synthesized CuO NPs are listed in Table 1, and are in good agreement with the standard data obtained from JCPDS card (No. 89-2529; *a* = 4.7 Å, *b* = 3.4 Å, *c* = 5.1 Å [20]). The lattice strains (*ε*) and average crystallite sizes (*D*) of the synthesized CuO were calculated using the following equations [20,21].
*Ε* = *β*/(4 tan*θ*)(1)
*D* = (0.9λ)/(*β*cos*θ*)(2)
where *β*, λ, and *θ* are the full width athalf maximum (FWHM) of the diffraction peak, the X-ray wavelength (1.5406 Å), and Bragg’s diffraction angle, respectively. The results listed in Table 1 indicate that the crystallite sizes of the synthesized CuO NPs are in the range of 8.3 nm to 15.8 nm, confirming the nanocrystalline nature of the synthesized CuO. The lattice strain values of the synthesized CuO NPs are in the range of 6.1 × 10^−3^ to 8.3 × 10^−3^, which are also close to those of the CuO NPs reported in literature (8.0 × 10^−3^) [21].

To further confirm the purity of the synthesized CuO NPs, their Fourier transform infrared (FT-IR) spectra were measured, and the results illustrated in Figure 3 and Appendix A in the Appendix A indicate that no FA, ML and BIM signals were observed in the FT-IR spectra of CuO NPs, which further confirms the purity of CuO NPs.

As illustrated in Figure 4 and Appendix A, the morphologies of the CuO NPs synthesized under different conditions are different; the CuO-1, CuO-2, CuO-3, CuO-4, CuO-7, and CuO-8 NPs are nanosheets, while flower-like shapes made up of stacked nanosheets can be observed for CuO-3, CuO-4, CuO-7, and CuO-8 NPs. Additionally, CuO-5 particles display a rod-like shape, and CuO-6 particles exhibit willow-leaf shaped nanosheets. These observations suggest that the morphologies of CuO are affected by the reaction temperature and NaOH concentration. Specially, higher reaction temperatures and NaOH concentrations tend to result in the formation of nano sheet-like CuO NPs. It should be noted that the CuO NPs prepared via the soft-chemical routes including solvothermal, ultrasound-assisted and microwave-assisted methods also exhibit leaf-like morphologies [1,2,4,6,8,17]. Their probable growth mechanism may be that in the synthesis process of CuO, Cu^2+^ ions react with OH^−^ to form Cu(OH)_2_ nuclei. These nuclei may then grow into layer structures and subsequently be converted into leaf-like CuO NPs through a dehydration process [4,6,8].

The optical properties of the synthesized CuO NPs were investigated via UV-Vis diffuse reflectance spectroscopy, and the results shown in Figure 5a and Appendix A in the Appendix A suggest that the synthesized CuO NPs have good absorption in the visible region; their band gap energy values (*E*_g_, Figure 5b and Appendix A) estimated using Tauc equation (Equation (3)) are in the range of 1.40 eV to 2.47 eV, and are consistent with the reported *E*_g_ values of CuO NPs [3,20,21,22,23].
(*a*h*v*)^2^ = K (h*v* − *E*_g_)(3)
where K, *E*_g_, *a*,and h*v* refer to the constant, the band gap energy, the absorption coefficient, and the photon energy, respectively.

Finally, the N_2_ adsorption–desorption isotherms of the synthesized CuO NPs were measured. The results, as illustrated in Figure 6 and Appendix A, suggest that all the synthesized CuO NPs show type-IV curves, which are characteristic of mesoporous materials with slit-like pores [24,25]. The specific surface areas (Brunauer–Emmett–Teller (BET) method) of the synthesized CuO NPs range from 12.4 m^2^ g^−1^ to 67.7 m^2^ g^−1^ (Table 1). The values are higher than those of commercially available CuO NPs (noted as CuO-9; BET specific surface area, 1.38 m^2^ g^−1^). It is notable that there is no direct correlation between the specific surface area and the catalytic activity of CuO NPs, as shown in Table 1 and Figure 7. For example, CuO-6 has the highest specific surface area (68 m^2^ g^−1^), but its catalytic activity is close to that of CuO-7 (specific surface area, 16 m^2^ g^−1^).

### 2.3. Photocatalytic Performance of the Synthesized CuO NPs

In this work, the photocatalytic performance of the synthesized CuO NPs was evaluated via the degradation of RhB in the presence of PMS. The results present in Figure 7 indicate that the CuO-2, CuO-3, CuO-4, CuO-6, and CuO-7 NPs exhibit higher photocatalytic activity in the degradation of RhB. Furthermore, all the synthesized CuO NPs show higher photocatalytic activity compared with the commercial CuO. Notably, CuO-7 NPs exhibit remarkable catalytic performance, achieving a degradation efficiency (*DE*) of 83.8% for RhB within 6.0 min. Therefore, the CuO-7 NPs were selected as photocatalysts in the subsequent experiments.

#### Effect of Parameters on RhB Degradation

The effect of various parameters, including the dosage of CuO, dosage of PMS, aqueous pH value and RhB concentration, on the RhB degradation was systematically studied. The results shown in Figure 8a indicate that without the addition of CuO, the *DE* of RhB is below 15%. However, in the presence of CuO, more than 96% of RhB can be degraded within 30 min, suggesting that PMS can be effectively activated by CuO NPs. As illustrated in Figure 8b, the *DE* of RhB increases with an increase in the PMS concentration. This may be attributed to the higher generation of reactive species resulting from a high PMS concentration [26].

As illustrated in Figure 9a, the CuO NPs maintain their high photocatalytic activity at pH 4 to pH 10. However, a remarkable decrease in the *DE* of RhB is observed at pH 12.0. This can be attributed to the fact that the pH point of zero charge (pHpzc) of CuO NPs is around 9.5, meaning that the charge on the CuO surface is negative at pH > 9.5 [27]. Consequently, at pH 12.0, the negative charge on the CuO surface repels the carboxylate group of RhB, thereby hindering its interaction with the CuO surface and leading to a lower *DE* of RhB [1]. Additionally, the decomposition of PMS is more pronounced instrong alkaline conditions (Equation (4)), leading to a reduction in the RhB degradation [26]. Since CuO NPs exhibit higher catalytic activity at pH 4 to pH 10, their stability was thus investigated, using pH 4 and pH 10 as examples. The experimental results suggest that the dissolution loss of CuO-7 NPs at pH 4 and pH 10 is 1.6% and 0.059%, respectively, indicating the good stability of CuO NPs.
HSO_5_^−^ + OH^−^→ SO_4_^2−^ + O_2_ + H_2_O(4)

The effect of the initial RhB concentration on the RhB degradation is illustrated in Figure 9b, and it can be seen that the *DE* of RhB decreases with the increase in the initial RhB concentration, which may be attributed to the fact that more active sites on CuO surface will be occupied at a high RhB concentration, resulting in an inefficient activation of PMS. Meanwhile, the degradation of RhB at a high concentration requires more reactive species, but no extra reactive species can be generated in the presence of the same dosage of PMS [27,28].

### 2.4. Identification of Reactive Species in the CuO-PMS System

Generally, hydroxyl radicals (•OH), sulfateradicals (•SO_4_^−^), singlet oxygen (^1^O_2_), superoxide radicals (•O_2_^−^) and photoexcited holes (h^+^) are the main reactive species in the CuO-PMS system [1,8,27], and are responsible for the oxidation of RhB. The generation of reactive species and the possible RhB degradation mechanism can be described by the following equations:CuO + photon (h*v*) → h^+^ + e^−^(5)
O_2_ + e^−^ → •O_2_^−^(6)
Cu^2+^ + HSO_5_^−^ → Cu^+^ + •SO_5_^−^ + H^+^(7)
Cu^+^ + HSO_5_^−^ → Cu^2+^ + •SO_4_^−^ + OH^−^(8)
Cu^+^ + HSO_5_^−^ → Cu^2+^ + SO_4_^2−^ + •OH(9)
2•SO_5_^−^ →^1^O_2_ + 2SO_4_^−^(10)
•SO_4_− + H_2_O → H^+^ + SO_4_^2−^ + •OH(11)
•O_2_^−^ + •OH → ^1^O_2_ + OH^−^(12)
•OH/•SO_4_−/^1^O_2_/•O_2_^−^/h^+^ + RhB → H_2_O + CO_2_ + …(13)

To identify the dominant reactive species generated in the CuO/PMS system, quenching experiments were carried out using methanol, *tert*-butanol (TBA), furfuryl alcohol (FFA), 1,4-benzoquinone (BQ), and ethylenediaminetetraacetic acid disodium salt (EDTA-2Na) as •SO_4_^−^, •OH, ^1^O_2_, •O_2_^−^, and h^+^ scavengers, respectively [1,8,27]. The results shown in Figure 10 indicate that the presence of a low concentration of EDTA-2Na (5.8 × 10^−3^ mol L^−1^) remarkably inhibits the RhB degradation, indicating that h^+^ can be effectively trapped by EDTA-2Na in the RhB degradation process, because EDTA-2Na is an electron donor and can interact with h^+^ via its lone-pair electrons [8].The inhibition effect of BQ on the RhB degradation is weaker than EDTA-2Na and stronger than FFA. The inhibition effects of TBA and methanol on the RhB degradation are rather poor compared with EDTA-2Na, BQ and FFA. These results suggest that the photoexcited holes, h^+^, are the predominant reactive species in the degradation of RhB; meanwhile, ^1^O_2_ and •O_2_^−^ also contribute to RhB degradation. Actually, Liu et al. also reported that photoexcited holes are the dominant reactive species for RhB degradation in the CuO/PMS system [8].

### 2.5. Reusability of the CuO NPs

The reusability of the synthesized CuO NPs was evaluated via recycling experiments. After each test, the used CuO NPs were collected from the suspension via centrifugation, washed with anhydrous ethanol, and dried at 80 °C for 4 h for reuse. The results shown in Figure 11 elucidate that no obvious loss in photocatalytic activity is observed after four cycles of the degradation test. Additionally, no obvious changes in structures were observed for the CuO NPs after four consecutive degradation cycles, as confirmed by the UV–Vis absorption spectra and XRD patterns (Figure 12). These observations suggest that the synthesized CuO NPs have good stability and reusability.

### 2.6. Comparison with Reported Catalysts

Recently, a variety of metal oxides/PMS systems were used to degrade RhB and a comparison of the RhB degradation performance of different catalysts was thus conducted. The results presented in Table 2 suggest that the synthesized mesoporous CuO NPs exhibit higher catalytic performance towards the RhB degradation compared with the reported catalysts; meanwhile, the RhB degradation reaction can be carried out under sunlight without the use of additional light sources, which is also one of the advantages of the synthesized CuO NPs.

## 3. Materials and Methods

### 3.1. Materials and Methods

The reagents, ML (99%), RhB (99%), Cu(NO_3_)_2_·3H_2_O (99%), PMS (2KHSO_5_·KHSO_4_·K_2_SO_4_), TBA (99%), BQ (99%) and spherical nano CuO (40 nm, 99.5%) were obtained from Macklin Biochemical Technology Co., Ltd. (Shanghai, China). FFA (98%), BIM (>98%), and FA (>96%) were supplied by Aladdin Biochemical Technology Co., Ltd. (Shanghai, China). All the other chemicals used are of analytical grade.

The hydrophobic NADES, ML-FA, was prepared by mixing ML with FA at a mole ratio of 1:1 at 70 °C until a homogeneous colorless liquid formed [18].

### 3.2. Synthesis of Mesoporous CuO NPs

Typically, 25 mL of 0.1 mol L^−1^ Cu(NO_3_)_2_ was mixed with 1.86 g of BIM under stirring at 25 °C to form a blue complex Cu(BIM)_4_(NO_3_)_2_ [19], followed by the addition of 10 mL ML-FA, leading to the transfer of Cu(BIM)_4_(NO_3_)_2_ from the water phase to the ML-FA phase. After removal of water phase, the ML-FA phase was contacted with 33 mL of 0.5 mol L^−1^ NaOH aqueous solution under stirring for 2 h at 80 °C to produce a black precipitate, which was then washed with water and ethanol, respectively, and dried at 70 °C for 4 h to obtain mesoporous CuO.

### 3.3. Characterization

The synthesized mesoporous CuO NPs were characterized via XRD (model X’Pert PRO MPD, PANalytical B.V., Almelo, The Netherlands), ultraviolet–visible (UV–Vis)/near-infrared (NIR) spectroscopy (Model UH4150, Hitachi High-Technologies Corp., Tokyo, Japan), and field emission scanning electron microscopy (FE-SEM, Quanta 250 FEG, Thermo Fisher Scientific, Hillsboro, OR, USA). The nitrogen gas adsorption–desorption isotherms at 77 K of the synthesized mesoporous CuO NPs were determined using an ASAP 2460 adsorption analyzer (Micromeritics Instrument Corp., Norcross, GA, USA).

### 3.4. Photocatalytic Activity of Mesoporous CuO NPs

The photocatalytic performance of the synthesized mesoporous CuO NPs was evaluated by degrading RhB in the presence of PMS. The degradation experiments were performed under sunlight irradiation at 25 °C with magnetic stirring. The pH values of the aqueous phase were adjusted using 0.1 mol L^−1^ HCl or NaOH solutions. For a typical experiment, 30 mL of RhB aqueous solution (20 mg L^−1^) was contacted with 5.0 mg of mesoporous CuO NPs under stirring at 25 °C for 10 min under dark to establish an adsorption–desorption equilibrium, and then 0.4 mL of PMS (10 g L^−1^) was added to the above solution; the resulting mixture was stirred immediately under sunlight irradiation, and 0.5 mL of the above mixture was withdrawn every 6 min and immediatelycentrifuged to remove the intervention of CuO NPs. The RhB concentrations were measured using a UV-Vis absorption spectrophotometer (model TU-1810, Purkinje General Instrument Co., Beijing, China) at 554 nm. The degradation efficiency (*DE*) of RhB was calculated using the following equation.
*DE*(%) = (1 − *C_t_*/*C*_0_) × 100(14) where *C*_0_ and *C_t_* are the initial RhB concentration and the RhB concentration in the degradation reaction carried out at *t* time, respectively.

### 3.5. Stability of CuO NPs

The stability of CuO NPs in water with different pH values (pH 4 and pH 10) was studied with CuO-7 NPs as a representative; 5.0 mg of CuO-7 NPs were mixed with 30 mL water (pH 4 or pH 10) for 24 h. The copper concentrations in water phase were measured via inductively coupled plasma optical emission spectroscopy (ICP-OES) (model Optima 8000, PerkinElmer Inc., Waltham, MA, USA). The dissolution loss of CuO was calculated using the following equation:Dissolution loss (%) = (mass of dissolved CuO)/(mass of original CuO) × 100(15)

## 4. Conclusions

In this work, mesoporous CuO NPs with a sheet-like structure were prepared in a green solvent system composed of a hydrophobic NADES and water under mild conditions. The morphologies and BET specific surface areas of the synthesized mesoporous CuO NPs were found to be influenced by the reaction temperature and NaOH concentration. The synthesized CuO NPs were applied to the degradation of RhB, and the experimental results suggested that the CuO/PMS system (0.16 g L^−1^ of CuO and 0.13 g L^−1^ of PMS) could degrade 98.0% of RhB (20 mg L^−1^) at pH 7.0 within 18 min. Photoexcited holes (h^+^) are the main reactive species generated in the CuO/PMS system. The reusability tests indicated that the CuO NPs could be reused at least four times without decreasing their catalytic ability significantly. Furthermore, the CuO NPs exhibited good stability in a pH range of 4 to 10. These findings indicated that CuO NPs were effective catalysts for the activation of PMS for the removal of organic pollutants.

## Figures and Tables

**Figure 1 molecules-28-05554-f001:**
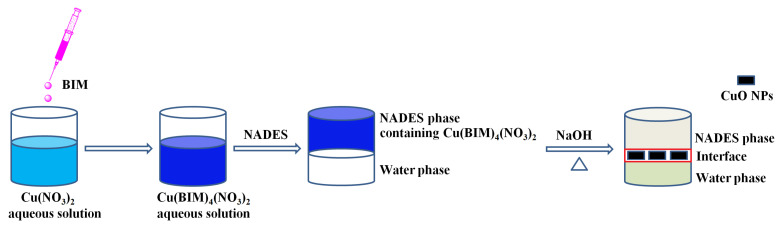
Schematic diagram of the preparation process of CuONPs in the NADES medium.

**Figure 2 molecules-28-05554-f002:**
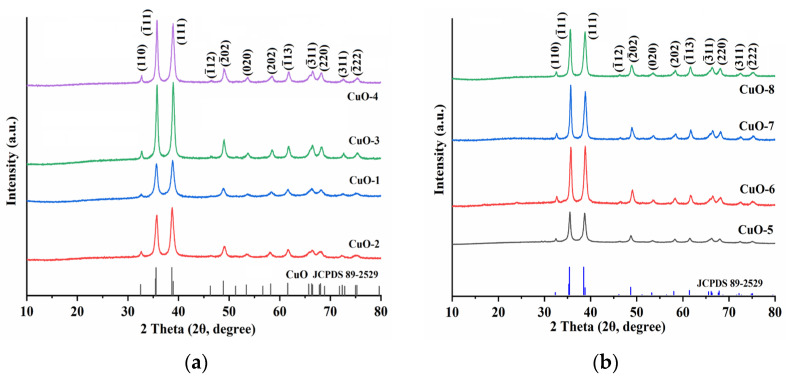
The XRD patterns of the synthesized CuO NPs; (**a**) CuO-1 to CuO-4 NPs; (**b**) CuO-5 to CuO-8 NPs.

**Figure 3 molecules-28-05554-f003:**
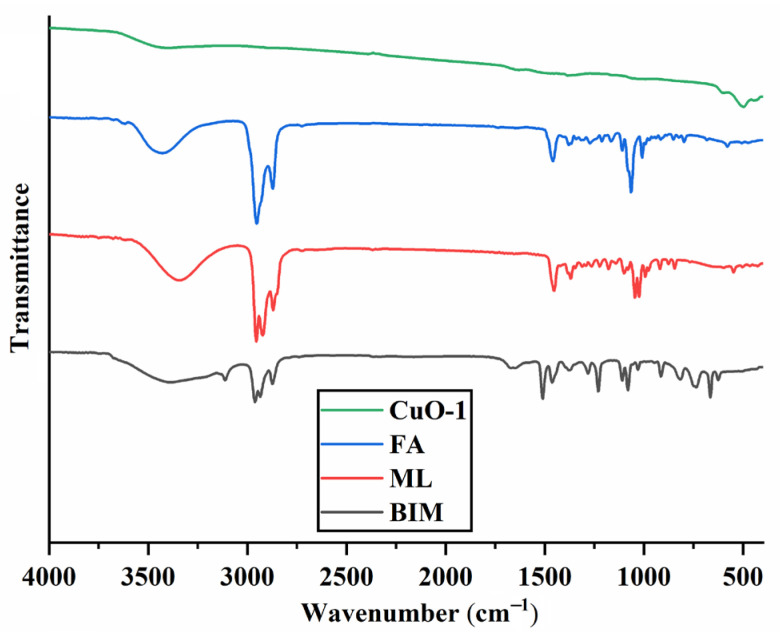
The FT-IR spectra of FA, ML, BIM and the synthesized mesoporous CuO-1 NPs.

**Figure 4 molecules-28-05554-f004:**
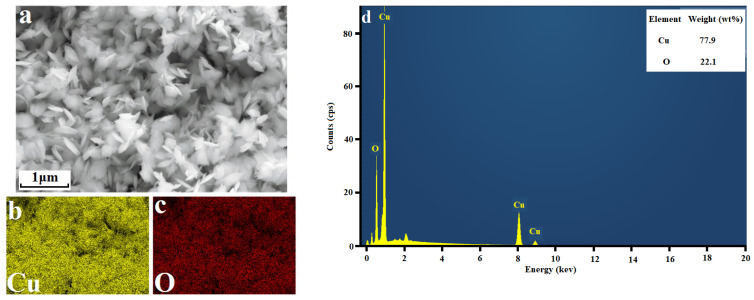
Scanning electron microscopic (SEM) image (**a**), energy-dispersive spectroscopic (EDS) mapping (**b**,**c**), and energy dispersive X-ray (EDX) pattern (**d**) of CuO-1 NPs.

**Figure 5 molecules-28-05554-f005:**
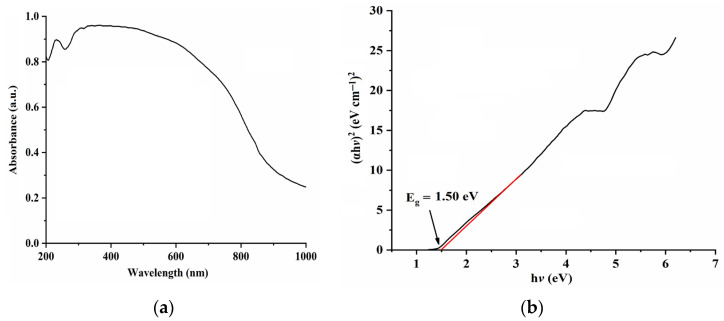
The UV–Vis absorption spectrum (**a**) and Tauc plot (**b**) of the synthesized mesoporous CuO-1 NPs. Red line in (**b**) is the tangent of Tauc curve.

**Figure 6 molecules-28-05554-f006:**
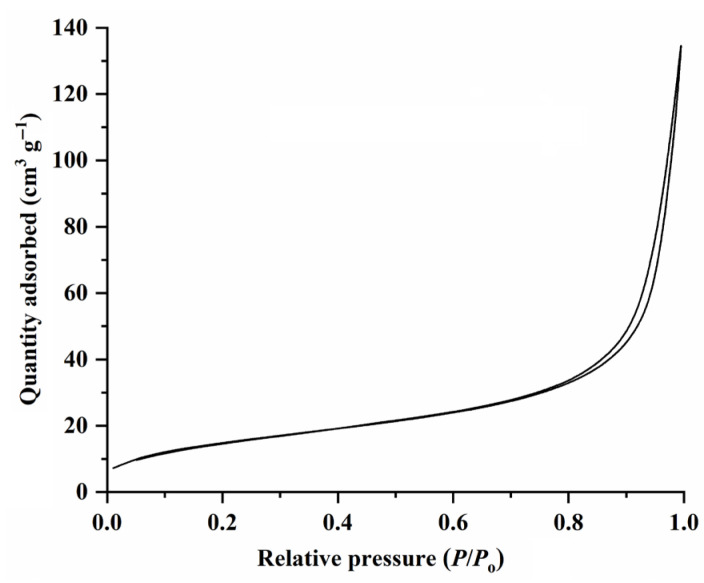
The N_2_ adsorption–desorption isotherms (77 K) of the synthesized CuO-1 NPs.

**Figure 7 molecules-28-05554-f007:**
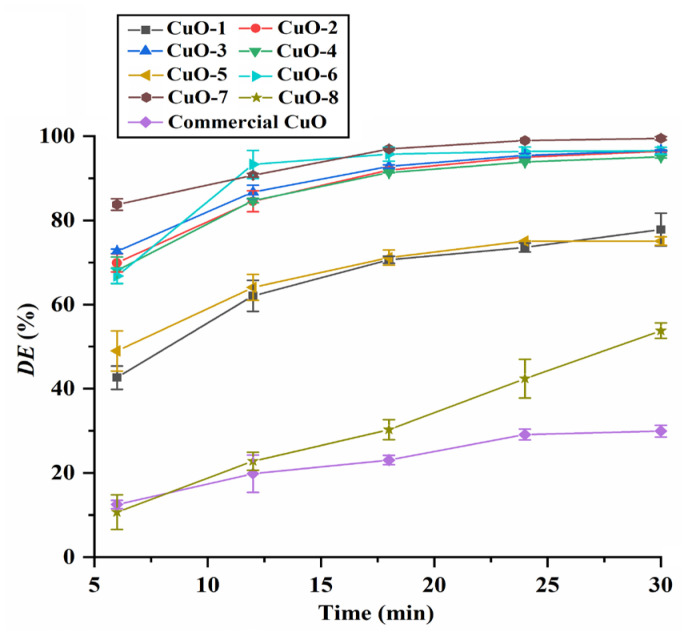
The efficiency (*DE*) of the synthesized CuO and commercial CuO NPs in degrading RhB. Experimental conditions: *C*_RhB_ = 20 mg L^−1^; *C*_CuO_ = 0.16 g L^−1^; *C*_PMS_ = 0.13 g L^−1^; pH 7.0.

**Figure 8 molecules-28-05554-f008:**
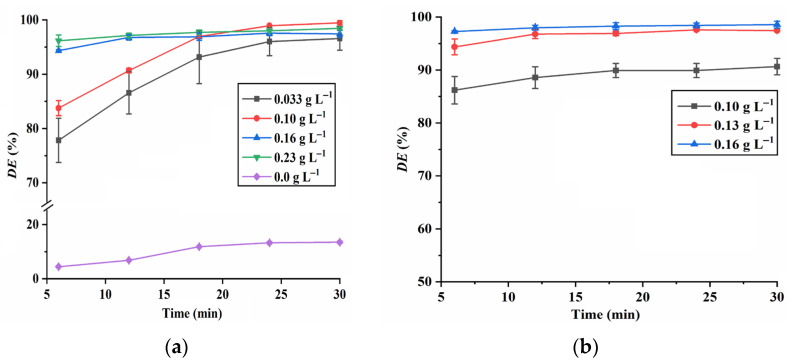
Effect of CuO dosage (**a**) (*C*_RhB_ = 20 mg L^−1^; *C*_PMS_ = 0.13 g L^−1^; pH 7.0) and PMS dosage (**b**) (*C*_RhB_ = 20 mg L^−1^; *C*_CuO_ = 0.16 g L^−1^; pH 7.0) on RhB degradation.

**Figure 9 molecules-28-05554-f009:**
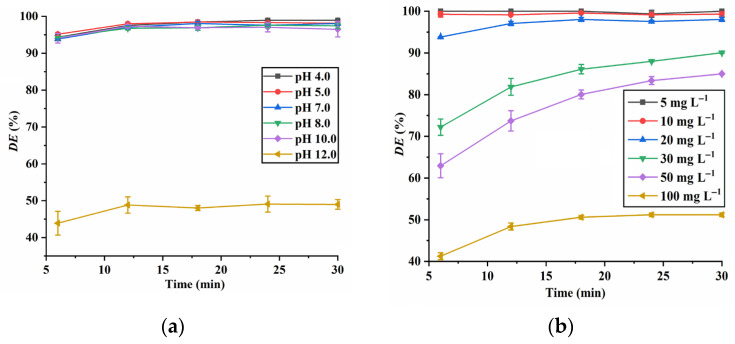
Effect of aqueous pH (**a**) (*C*_RhB_ = 20 mg L^−1^; *C*_CuO_ = 0.16 g L^−1^; *C*_PMS_ = 0.13 g L^−1^) and initial RhB concentration (**b**) (*C*_CuO_ = 0.16 g L^−1^; *C*_PMS_ = 0.13 g L^−1^; pH 7.0) on the RhB degradation.

**Figure 10 molecules-28-05554-f010:**
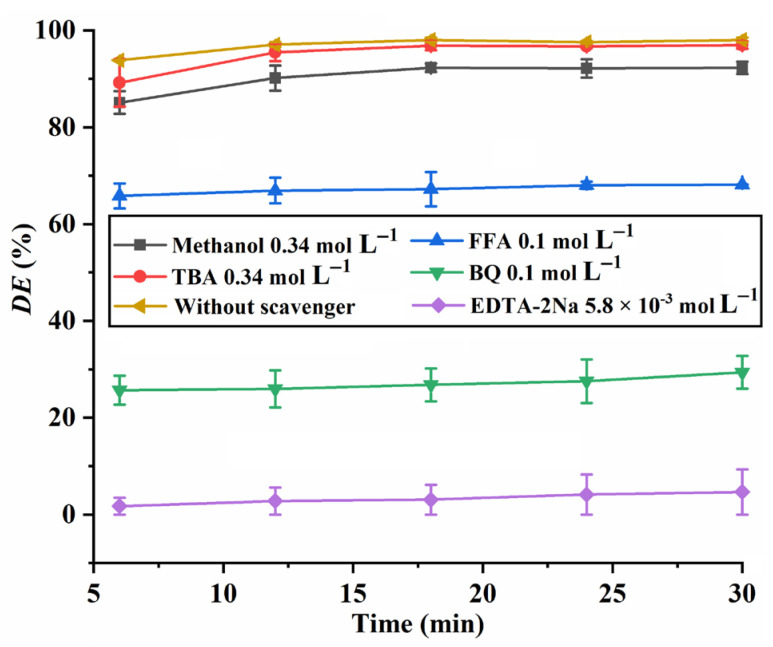
The *DE* of RhB with and without the presence of scavengers. Experimental conditions: *C*_RhB_ = 20 mg L^−1^; *C*_CuO_ = 0.16 g L^−1^; *C*_PMS_ = 0.13 g L^−1^; pH 7.0.

**Figure 11 molecules-28-05554-f011:**
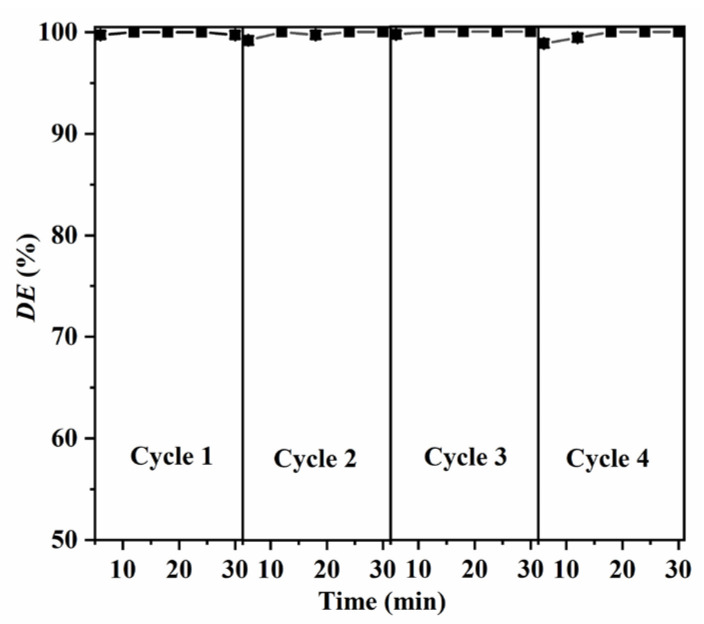
Reusability of the synthesizedCuO-7 NPs. Experimental conditions: *C*_RhB_ = 10 mg L^−1^; *C*_CuO_ = 0.16 g L^−1^; *C*_PMS_ = 0.13 g L^−1^; pH 7.0.

**Figure 12 molecules-28-05554-f012:**
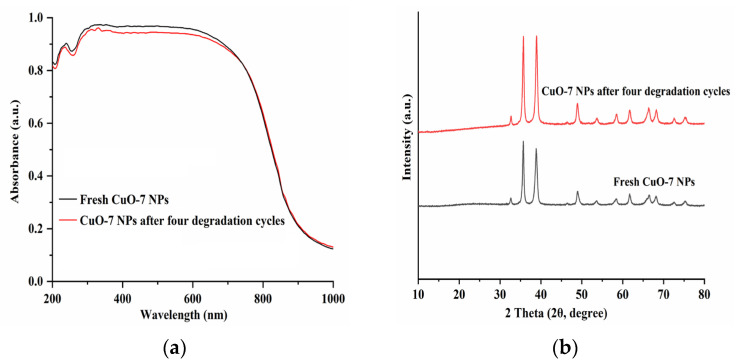
The UV–Vis absorption spectra (**a**) and XRD patterns (**b**) of the fresh CuO-7and CuO-7 after four degradation cycles. Experimental conditions: *C*_RhB_ = 10 mg L^−1^; *C*_CuO_ = 0.16 g L^−1^; *C*_PMS_ = 0.13 g L^−1^; pH 7.0.

**Table 1 molecules-28-05554-t001:** The specific surface areas (Brunauer–Emmett–Teller (BET) method), pore volumes, pore sizes, crystallite sizes, lattice strains and lattice parameters of the synthesized CuO NPs.

Sample No.	Preparation Conditions	BET Specific Surface Area (m^2^ g^−1^)	Pore Volume (cm^3^ g^−1^)	Pore Size (nm)	Crystallite Size (nm)	Lattice Strain	Lattice Parameter (Å)
CuO-1	0.05 mol L^−1^ NaOH; reaction temperature, 60 °C; 2 h of reaction time under stirring	54	9.3 × 10^−2^	9.8	8.3	8.1 × 10^−3^	*a* = 4.7, *b* = 3.4, *c* = 5.1
CuO-2	0.1 mol L^−1^ NaOH; reaction temperature, 60 °C; 2 h of reaction time under stirring	41	8.3 × 10^−2^	21	9.1	8.3 × 10^−3^	*a* = 4.7, *b* = 3.4, *c* = 5.1
CuO-3	0.5 mol L^−1^ NaOH; reaction temperature, 60 °C; 2 h of reaction time under stirring	21	4.2 × 10^−2^	11	15	6.6 × 10^−3^	*a* = 4.7, *b* = 3.4, *c* = 5.1
CuO-4	1.0 mol L^−1^ NaOH; reaction temperature, 60 °C; 2 h of reaction time under stirring	17	3.8 × 10^−2^	21	14	6.9 × 10^−3^	*a* = 4.7, *b* = 3.4, *c* = 5.1
CuO-5	0.5 mol L^−1^ NaOH; reaction temperature, 25 °C; 2 h of reaction time under stirring	40	6.8 × 10^−2^	16	11	7.0 × 10^−3^	*a* = 4.7, *b* = 3.4, *c* = 5.1
CuO-6	0.5 mol L^−1^ NaOH; reaction temperature, 40 °C; 2 h of reaction time under stirring	68	1.3 × 10^−1^	20	11	6.5 × 10^−3^	*a* = 4.7, *b* = 3.4, *c* = 5.1
CuO-7	0.5 mol L^−1^ NaOH; reaction temperature, 80 °C; 2 h of reaction time under stirring	16	3.5 × 10^−2^	23	16	6.1 × 10^−3^	*a* = 4.7, *b* = 3.4, *c* = 5.1
CuO-8	0.5 mol L^−1^ NaOH; reaction temperature, 100 °C; 2 h of reaction time under stirring	12	2.9 × 10^−2^	12	15	6.8 × 10^−3^	*a* = 4.7, *b* = 3.4, *c* = 5.1

**Table 2 molecules-28-05554-t002:** Comparison of different catalysts on the RhB degradation.

Catalyst	Synthesis Method	Experimental Conditions	References
CuO	NADES/water-based two-phase interface method (80 °C for 2 h under stirring)	RhB concentration, 20 mg L^−1^ (pH 7); 0.13 g L^−1^ PMS; 0.16 g L^−1^ CuO; degradation time, 18 min; *DE*, 98.0%; light source, sunlight	This work
CuO	Precipitation method (Cu^2+^ + sodium citrate + NaOH; 80 °C for 20 min, followed by 100 °C for 10 min under stirring)	RhB concentration, 20 mg L^−1^ (pH 7); 0.12 g L^−1^ PMS; 0.3 g L^−1^ CuO; degradation time, 20 min; *DE*, 98.0%; light source, a 300 W Xe lamp with a UV cut-off filter	[8]
*α*-MnO_2_	hydrothermal method (140 °C for 12 h)	RhB concentration, 20 mg L^−1^;0.20 g L^−1^ PMS; 0.10 g L^−1^α-MnO_2_; degradation time, 60 min; *DE*, 99%; light source, sunlight	[28]
CuO-CeO_2_	calcination method (550 °C for 4 h)	RhB concentration, 47.9 mg L^−1^ (pH 7);0.98 g L^−1^ PMS; 0.4 g L^−1^CuO-CeO_2_; degradation time, 60 min; *DE*, 100%; light source, sunlight	[29]
Conjugated polyvinyl chloride (cPVC)/Bi_2_O_3_	hydrothermalcalcination method: 160 °C for 3 h for the hydrothermal procedure and 350 °C for 2 h for the calcination procedure	RhB concentration, 200 mg L^−1^; 1.8 g L^−1^ PMS; 0.33 g L^−1^ catalyst; degradation time, 150 min; *DE*, 100%; light source, simulated solar light (300 W) witha UV cuttingglass	[30]
Co_3_O_4_–rice husk ash composites	hydrothermalcalcination method: 120 °C for 6 h for the hydrothermal procedure and 350 °C for 2 h for the calcination procedure	RhB concentration, 20 mg L^−1^ (pH 6); 0.5 g L^−1^ PMS; 0.1 g L^−1^ catalyst; degradation time, 60 min; *DE*, 96.3%; light source, sunlight	[31]

## Data Availability

Data are contained within the article.

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
