# Peer review of "Mesoporous CuO Prepared in a Natural Deep Eutectic Solvent Medium for Effective Photodegradation of Rhodamine B"

_molecules, 2023, doi:10.3390/molecules28145554_

Round 1
Reviewer 1 Report
The manuscript “Mesoporous CuO Prepared in Natural Deep Eutectic Solvent Medium for Effective Photodegradation of Rhodamine B” is an interesting piece of work as it presents the synthesis of inorganic oxide in hydrophobic medium. Nevertheless, the design of the experiment and the results are severely underdiscussed. Below I list some questions that arose during the reading of the manuscript.
There is a wide literature concerning the high toxicity of copper compounds including CuO with respect to various microorganisms and water-related fauna. E.g., CuO is widely used as an additive to antifouling dyes. In this respect, the low toxicity of CuO is highly questionable and can not be regarded as a pro for the material.
Generally, can the menthol-furfuryl alcohol mixture be even regarded as DES? I think the discussion of this issue is of primary importance for the paper. As far as I know, only methol-fatty acid mixtures were reported as DESs until now.
In a literature survey, there is no need to present the exact details of CuO synthesis by other authors.
Recently, CuO synthesis in DESs has been reported, see e.g. 10.1016/j.molliq.2021.117319 (for the Editor’s concern: here, I declare no conflict of interest). I am almost sure that there are some more examples. The previous efforts to synthesise CuO using DESs must be mentioned and discussed.
From the experimental section, I see that 10 mL of the solution of Cu(BIM)4(NO3)2 in menthol-furfuryl alcohol mixture was mixed with 33 mL of NaOH aqueous solution. I have some concerns and questions on the process details that were not disclosed in the paper. Where the formation of CuO NPs took place, on the liquid interface? Are the components of menthol-furfuryl alcohol mixture miscible with water (furfuryl alcohol is miscible), was it checked? Is the hydrolysis of furfuryl alcohol possible under experimental conditions (it can react with water)? Can BIM transfer to menthol-furfuryl alcohol mixture from aq. phase (if so, the DES is not a two- but three-component), was it checked?
CuO particles were obtained as leafs. This is an interesting result. Is the leaflike morphology typical for CuO particles synthesised by soft-chemical routes? The comparison of the morphology with different methods must be provided. The reasons for such morphology must be discussed.
Please keep in mind the low accuracy of low temperature nitrogen adsorption measurement results, and remove insignificant digits from the numerical values.
Is CuO stable at pH 4, pH 10? Can it dissolve? This issue must be checked.
Fig. 8 presents the kinetic analysis. Here I see poor linear regressions that can not reach R 0.99. Please re-check the correlation coefficients. From such poor linearity, I recommend the authors to avoid making any strong conclusions.
I strongly recommend the authors to provide IR-measurements in the samples to check the adsorption of BIM, methol, FA on the surface of the particles. Is the content of organic matter on the particles the same for various samples? Can this organic matter influence the photocatalytic activity of the samples?
The direct comparison of the photocatalytic activity of the materials (Table 2) is incorrect as the corresponding results were obtained under brand different experimental conditions (lamps, volumes, concentrations, pH). Generally, the claim “These results suggested that NADESs may be ideal media for the preparation of metal oxide nanoparticles with excellent catalytic activity” is highly questionable as the comparison with other methods and media was conducted improperly.
Are the CuO aq. suspensions stable? Do they sediment?
Please, check the abbreviations FFA and FA. Do they attribute to the same substance?
In Fig. 6, please bring the both panels to the same Y-coordinates. You can use axis break in panel (a). This can help to compare the curves in panels (a) and (b).
The authors claim that “Photoexcited holes (h+) are the main reactive species generated in the CuO/PMS system”. Is this point typical to CuO photocatalysts?
Taking in mind the above points, I recommend a major revision though the paper is somehow closer to the reject decision. In order to address all my concerns, the manuscript must be complemented with new experimental data and their discussion. I hope the authors will succeed in such a challenge.
The English is acceptable, but some spellcheck is needed.
Reviewer 2 Report
In this manuscript, the authors reported "Mesoporous CuO Prepared in Natural Deep Eutectic Solvent Medium for Effective Photodegradation of Rhodamine B". The manuscript is good but needs to improve. I recommend it be accepted for publication after major revision. The main concerns are as follows.
1. The authors need to change the title of the manuscript.
2. The abstract should be written more precisely without including unnecessary information. Try to highlight the novelty of the research and its contribution accurately.
3. In the abstract: line numbers: 10 & 16 check it carefully.
4. Rewrite the keywords.
5. In keywords: The authors had already abbreviated NADEs & RhB previously; then why authors did again abbreviate in this section?
6. The introduction section needs to be rewritten to incorporate recent work and emphasize the significance of using CuO nanoparticles. The authors should provide a comprehensive overview of the field and explain how these particular CuO nanoparticles for RhB degradation address current research gaps or challenges.
7. In Figure 1: The authors should index the observed XRD peaks.
8. From Figure 1; the authors should calculate the structural parameters such as lattice constant, crystallite size, strain, etc., of the prepared samples.
9. The authors should add the calculated structural parameters discussion in the structural analysis section.
10. The authors need to include the EDAX analysis of the optimized catalysts.
11. In Figure 3a & b: Include the y-axis unit.
12. In the supporting information; Figure S1-S7: Include the y-axis unit.
13. The authors need to shift Table 1 after Figure 4.
14. In Figures 5a & b: The authors should combine the degradation efficiency in one graph.
15. The authors should improve the photocatalytic performance of the synthesized CuO nanoparticles discussion using BET analysis.
16. In Figure 8: the "r" or "r2"?? Check it.
17. Why EDTA-2Na scavenger is more capable of inhibiting RhB degradation compared to other scavengers? Explain it.
18. Post-photocatalysis characterizations are required; e.g. XRD or UV-vis to further prove the stability of the CuO nanoparticles.
19. The authors must include the plausible photocatalytic RhB degradation mechanism using prepared CuO nanoparticles.
20. In Table 2: The authors need to include the synthesis method and light source.
21. The authors should include the novelty in the conclusion part.
22. There are more typos in the manuscript; double-check it thoroughly.
23. Improve the figure captions and add more details.
24. The authors should polish the English carefully and thoroughly in the manuscript.
In summary, the manuscript requires modification in the title, abstract, introduction, and results & discussion sections to address the points mentioned above. By incorporating these suggestions, the manuscript will provide a clear understanding of the research and its implications in the field of photocatalysts.
The authors should polish the English carefully and thoroughly in the manuscript.
Round 2
Reviewer 1 Report
The authors cleared most of my concerns, some minor issues remained.
1) I asked to remove insignificant digits, not insignificant values. In the manuscript, the surface area values, crystallite sizes are still given with an excess accuracy. Please check and correct.
2) I am still concerned on the correlation coefficients in the kinetic data. The regression in Fig. 10a can not result in 0.99. I checked and obtained only 0.91. In this regard, taking in mind very poor linearization of the kinetic data in both models, I see no sense in discussion the kinetics of the reaction at all.
3) As for the title. The title was changed to “Natural Deep Eutectic Solvents: Ideal Media for Preparation of CuO with Enhanced Photocatalytic Activity”. As for me, the previous title was better as it was neutral. The new one is too pretentious. From the manuscript, I do not see any serious evidences that DESs are ideal media. They are ideal economically? No. Provide the highest _possible_ photocatalytic activity? No, too. Thus, I do not recommend using the word “ideal” or other similar words for the title.
English is acceptable.
Reviewer 2 Report
The authors have responded to the reviewer's comments and can be accepted.
Author Response
Thank you very much for your comments!